# Large Language Models Are Semi-Parametric Reinforcement Learning Agents

**Danyang Zhang**[1]  **Lu Chen**[1,2][†]  **Situo Zhang**[1]

**Hongshen Xu**[1]  **Zihan Zhao**[1]  **Kai Yu**[1,2]

[1]X-LANCE Lab, Department of Computer Science and Engineering
MoE Key Lab of Artificial Intelligence, SJTU AI Institute
Shanghai Jiao Tong University, Shanghai, China
[2]Suzhou Laboratory, Suzhou, China
{zhang-dy20,chenlusz,situozhang,xuhongshen,zhao_mengxin,kai.yu}@sjtu.edu.cn

## Abstract

Inspired by the insights in cognitive science with respect to human memory and reasoning mechanism, a novel evolvable LLM-based (Large Language Model) agent framework is proposed as REMEMBERER. By equipping the LLM with a long-term experience memory, REMEMBERER is capable of exploiting the experiences from the past episodes even for different task goals, which excels an LLM-based agent with fixed exemplars or equipped with a transient working memory. We further introduce **R**einforcement **L**earning with **E**xperience **M**emory (**RLEM**) to update the memory. Thus, the whole system can learn from the experiences of both success and failure, and evolve its capability without fine-tuning the parameters of the LLM. In this way, the proposed REMEMBERER constitutes a semi-parametric RL agent. Extensive experiments are conducted on two RL task sets to evaluate the proposed framework. The average results with different initialization and training sets exceed the prior SOTA by 4% and 2% for the success rate on two task sets and demonstrate the superiority and robustness of REMEMBERER.[1]

## 1 Introduction

*Reasoning is remembering.* As declared by Seifert et al. [1997], the episodic memory of the experiences from past episodes plays a crucial role in the complex decision-making processes of human [Suddendorf and Corballis, 2007]. By recollecting the experiences from past episodes, the human can learn from success to repeat it and learn from failure to avoid it. Similarly, an agent should optimize its policy for a decision-making task with the help of reminiscence of the interaction experiences. In this work, we primarily investigate how to utilize large language models (LLMs) as agents and equip them with interaction experiences to solve sequential decision-making tasks.

Despite the impressive performance of LLMs on many natural language processing (NLP) tasks [Wei et al., 2022, Kojima et al., 2022, Wang et al., 2022, Yao et al., 2022b], existing approaches still struggle to enable LLMs to effectively learn from interaction experiences. On the one hand, the most common approach for an agent to utilize the experiences is to fine-tune the model parameters through reinforcement learning (RL). However, it requires a considerable expenditure to deploy and fine-tune an LLM, which makes it difficult to apply task-aware RL to the LLM to preserve the experiences. On the other hand, recent work like Algorithm Distillation [Laskin et al., 2022] presents an in-context

---

[†]Lu Chen is the corresponding author.
[1]The codes are open-sourced at `https://github.com/OpenDFM/Rememberer`.

37th Conference on Neural Information Processing Systems (NeurIPS 2023).

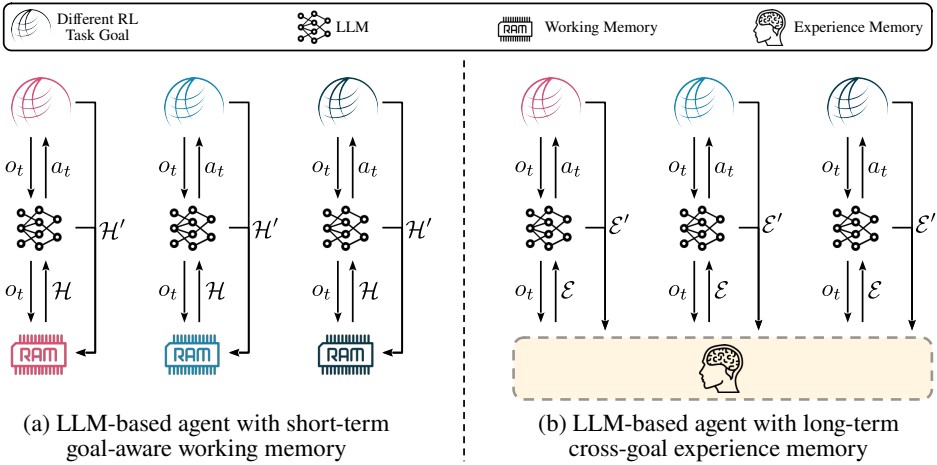

Figure 1: Comparison of the LLM-based agents with short-term working memory and long-term experience memory. The working memory stores only the historical information of the current episode ($\mathcal{H}$). while the experience memory stores the interaction experiences ($\mathcal{E}$) permanently.

reinforcement learning by embedding the RL training trajectories into the input prompt of a pretrained decision transformer. This method manages to make use of past interaction experiences without model fine-tuning. However, existing LLMs suffer from a serious limitation of the input length to embed the whole experience. Hence, to better store a plethora of interaction histories and aid LLMs in learning during the interaction process, we introduce **RLEM**, *i.e.*, **R**einforcement **L**earning with **E**xperience **M**emory, which accomplishes agent learning by updating the experience memory through the RL process, rather than modifying the model parameters.

An external experience memory is different from the existing work like Reflexion [Shinn et al., 2023] which combines the LLM with a short-term working memory. As depicted in Figure 1 (a), a working memory is tied to a specific task goal, and the stored histories cannot be leveraged in future episodes for different goals. This analogy can be drawn to the Random Access Memory (RAM) of a computer, where stored information is lost in the event of power removal. On the other side, learning from the successful or failed experiences stored in the memory is different from the existing work like Inner Monologue [Huang et al., 2022b], Corrective Re-Prompting [Raman et al., 2022], and DEPS [Wang et al., 2023b] that takes advantage of immediate failure feedback only once. Storing long-term experiences in a persistent memory gives an opportunity to discover the late failure and learn from the experiences in the past episodes even for different task goals (see Figure 1 (b)).

By combining RLEM with LLM, we propose Rememberer, an evolvable LLM-based agent framework for decision-making tasks. Remememberer can utilize the experiences stored in the memory selectively in accordance with the current interaction state to optimize the decision. Meanwhile, the experience memory can be updated through an RL process constantly. Such a joint system is regarded as a semi-parametric RL agent, which can evolve its ability through its interaction experiences analogically to a full-parametric system, however, without fine-tuning the LLM parameters. We evaluate Remememberer on two recent RL task sets with the promising performance of LLM-based agents, WebShop [Yao et al., 2022a] and WikiHow [Zhang et al., 2023]. The agent is trained on a few tasks and tested on some other tasks to check whether the experiences from different tasks can help the agent in the decision of the unseen episodes. Remememberer demonstrates a significant performance boost compared to both previous SOTA and our fixed-exemplar LLM baselines. Specifically, it achieves an average improvement of 2 points and 4 points on the Webshop and WikiHow tasks, respectively, compared to the SOTA models.

Our contributions are summarized as follows: 1) A new agent framework is proposed as Rememberer for LLM to learn from the experiences of past episodes. The experiences are stored in an external persistent memory instead of fine-tuning the LLM parameters or forming an extremely long prompt. 2) We introduce RLEM, which updates experience memory through analogical RL training so that Rememberer is capable of self-evolving. 3) Rememberer manages to bypass the baseline

and the prior advanced performances and set up a new state of the art on two recent benchmarks, WebShop (+2 points on SOTA) and WikiHow (+4 points on SOTA).

## 2 Related work

**LLM with external information** External information is usually adopted to augment the LLM with the environment-grounded information, or to reduce the hallucination, or to unleash the ability to process longer context. Connecting with an external knowledge base is a common choice for question-answering and conversational tasks [Peng et al., 2023, Schick et al., 2023, Trivedi et al., 2022, Pan et al., 2022]. However, an external knowledge base is usually not directly corresponding to an RL environment and cannot provide environment-grounded assistance to an automatic agent. Meanwhile, the update to a mature knowledge base may not be in time for the instant interaction of the agent with the environment. In contrast, Schuurmans [2023] simulates a universal Turing machine with a RAM-augmented LLM and demonstrates the capability of a quickly-updatable working memory. Liang et al. [2023] and Zhong et al. [2023] adopt memory to store the conversational history and handle extremely long contexts. Relational database is leveraged to track states in a dynamic process by ChatDB [Hu et al., 2023]. Reflexion [Shinn et al., 2023] exploits a working memory to store experiences for a dedicated task to improve the performance of the agent through several trials. However, as illustrated in Figure 1, the histories stored in working memory cannot benefit the episode for different task goals. Instead, a long-term cross-goal experience memory should be considered. MemPrompt [Madaan et al., 2022] and Ret-LLM [Modarressi et al., 2023] adopt a persistent memory to store human feedback and remind the chatbot of the conversational knowledge and improve it continuously. Voyager [Wang et al., 2023a] designs a skill library to store the past learned skills as JavaScript functions. A simple text experience pool is adopted by GITM [Zhu et al., 2023] to store the successful trajectories for future referencing. Somewhat similar to GITM, REMEMBERER adopts a persistent environment-grounded experience memory to store the experiences and assist in future decision-making even for different task goal. However, instead of plain text records of successful trajectories, REMEMBERER uses a structured memory and designs a mechanism to task advantage of both successful and failed experiences. The experiences come from the interaction of the agent with the environment, and no human intervention is needed.

**LLM learning from failure** Learning from failure is one of the characteristic capabilities of human and turns to be an important topic for general artificial intelligence. Some work has explored the ability of the LLM to learn from its failure [Huang et al., 2022b, Raman et al., 2022, Wang et al., 2023b]. Nonetheless, most of such work takes advantage of immediate feedback from the environment and the correction is used only once. In practice, several late failures may be due to some early mistaken actions in an episode. Reflexion [Shinn et al., 2023] designs a heuristic function to detect late failure from the interaction history and stores the LLM-generated reflection in a working memory for use in the next trial. However, these reflections cannot be applied to different task goals. Madaan et al. [2022] stores the failure corrections for a long term, but relies on human feedback. In contrast, REMEMBERER adopts RL to learn from both late and immediate failure from the environment rewards without need for human feedback. Also, REMEMBERER enables the experiences to be reused in the future episode even for a different task goal with a long-term experience memory.

**LLM for decision-making** The powerful capability of LLM is exploited by recent work [Huang et al., 2022a, Raman et al., 2022, Mees et al., 2022, Chen et al., 2022, Ichter et al., 2022, Huang et al., 2022b, Liang et al., 2022] to generate better control plans for various robots and agents. Kim et al. [2023] and Zhang et al. [2023] design LLM-based agents for user interface (UI) interaction. ReAct [Yao et al., 2022b] combines the action decision with natural language reasoning and achieves a promising performance. To our best knowledge, This work is the first one that combines the LLM-based agent with RL algorithm to learn from the interaction experiences and achieve self-evolving.

The proposed REMEMBERER equips the LLM with an external experience memory to help it to learn from both successful and failed experiences. This is also the first work to combine the LLM-based agent with RL algorithm to improve the capability of the agent.

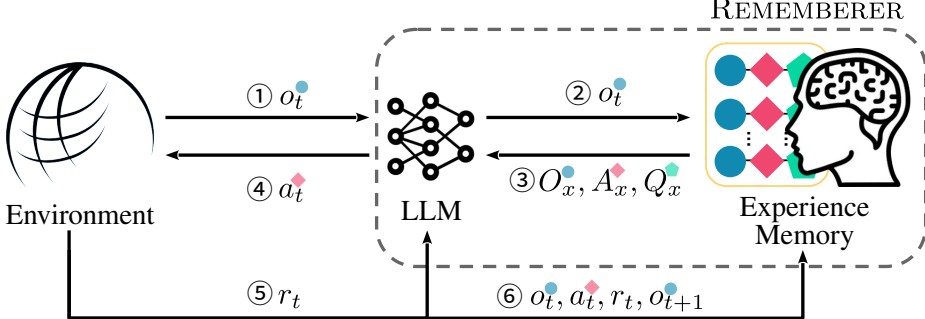

Figure 2: Pipeline of RLEM and architecture of REMEMBERER

## 3  Method

### 3.1  RLEM pipeline

RLEM (Reinforcement Learning with Experience Memory) is proposed for an LLM-based agent to learn from its interaction experiences by updating an external persistent memory. The pipeline of RLEM and the architecture of REMEMBERER agent are depicted in Figure 2. REMEMBERER agent consists of two components: an LLM making decisions and an experience memory storing the interaction experiences. At the decision step, the LLM first takes an observation $o_t$ from the environment. The observation $o_t$ is then adopted to retrieve several related experiences from the connected experience memory according to some similarity functions. The experiences are represented as a group of observations $O_x$, actions $A_x$, and the corresponding $Q$ value estimations $Q_x$. Here $x$ denotes the index set of retrieved experiences and depends on the specific runtime observation $o_t$. Subsequently, LLM will decide the action $a_t$ in accordance with $o_t$, the feedback from the last interaction (*e.g.*, the reward $r_{t-1}$), as well as the retrieved experiences $(O_x, A_x, Q_x)$. $a_t$ will be executed in the environment and the resulted reward $r_t$ will be returned to the LLM as the feedback. And the transition tuple, $(o_t, a_t, r_t, o_{t+1})$, comprising the last observation, the taken action, the corresponding reward, and the new observation will be used to update the experience memory. The following subsections will detail the structure and updating policy of REMEMBERER experience memory and the usage of the retrieved experiences.

### 3.2  Experience memory of REMEMBERER

The experience memory is one of the pivotal components of the proposed REMEMBERER framework. It is adopted to store the interaction experiences, and the LLM is expected to benefit from the stored experiences in future decision-making. The memory can be regarded as a group of external parameters of the LLM-based agent. Such an agent is a semi-parametric system that can evolve through RL process. During the interaction, new experiences are added to the experience memory so that the overall system can attain a more capable interaction ability compared to the agents with just a fixed LLM

| Task & Obsv. | Action | Q Value |
|:---:|:---:|:---:|
| $(g_1, o_1)$ | $a_1$ | $q_1$ |
| $(g_2, o_2)$ | $a_2$ | $q_2$ |
| $(g_3, o_3)$ | $a_3$ | $q_3$ |
| $\vdots$ | $\vdots$ | $\vdots$ |

Figure 3: An example of the records stored in the proposed experience memory.

and fixed exemplars. This procedure can be considered analogous to the training stage of conventional parametric agents.

To be specific, the experience memory is designed as a table storing the task information, observation, action, and the corresponding $Q$ value estimation. The $Q$ value is the expectation of the accumulated future reward and gives an assessment of the value of the action candidate. Figure 3 depicts a demonstration of the proposed experience memory. There are two stages to build a practical REMEMBERER agent with experience memory: *initialization* and *training*. The experience memory is supposed to be first initialized with some initial records before the training stage. The initial

records are necessary to inform the LLM of the format of the input and the output. Then, during the analogical training stage, the agent interacts with the environment to collect new experiences, and conducts off-policy learning [Sutton and Barto, 1999]. Particularly, given the task information $g$ and the new transition $(o_t, a_t, r_t, o_{t+1})$, as a quadruple of the last observation, action, reward, and the new observation, a new estimation is calculated first according to the estimated Bellman optimality equation [Bellman, 1952] as

$$Q'(g, o_t, a_t) = r_t + \gamma \max_a Q(g, o_{t+1}, a). \tag{1}$$

Here $\max$ can be calculated from the actions already recorded for $(g, o_{t+1})$ by considering the $Q$ value of unrecorded actions 0, if the action space cannot be traversed, *e.g.*, action space involving free-form language. Then a new record is inserted directly if there does not exist a record associated to $(g, o_t, a_t)$ in the memory:

$$Q(g, o_t, a_t) = Q'(g, o_t, a_t). \tag{2}$$

If $(g, o_t, a_t)$ has been already inserted into the record, the recorded $Q$ value estimation will be updated by Q-Learning [Watkins and Dayan, 1992]:

$$Q(g, o_t, a_t) \leftarrow (1 - \alpha)Q(g, o_t, a_t) + \alpha Q'(g, o_t, a_t). \tag{3}$$

Here the learning rate, $\alpha$, is $1/N$ where $N$ denotes the times this value is updated. As Equation 1 may lead to an inaccurate estimation owing to insufficient sampling out of few training steps of REMEMBERER, $n$-step bootstrapping [Mnih et al., 2016] is adopted to ameliorate this problem, which estimates $Q'$ by

$$Q'(g, o_t, a_t) = \sum_{i=0}^{n-1} \gamma^i r_{t+i} + \gamma^n \max_a Q(g, o_{t+n}, a), \tag{4}$$

where $n$ is the steps to expand. The ablation study in Subsection 4.4 proves this perspective.

### 3.3 Usage of the experiences

In order to assist the LLM in making decisions, the stored experiences are adopted as dynamic exemplars for few-shot in-context learning. Given the task goal $g$ and the current observation $o_t$, a similarity function $f$ is used to calculate the similarity of $(g, o_t)$ with $(g_i, o_i)$ from the memory.

$$S_i = f((g, o_t), (g_i, o_i)). \tag{5}$$

Commonly, a similarity function $f$ can be divided into two components, task similarity $f_g$ and observation similarity $f_o$:

$$S_i = \lambda f_g(g, g_i) + (1 - \lambda)f_o(o_t, o_i). \tag{6}$$

The $m$ records with the highest similarities are retrieved to form the exemplars in the prompt. The particular similarity function designed for each task set is detailed in Subsection 4.1.

The exemplar is supposed to demonstrate the format of the input and the output to the LLM. The input part usually comprises the task information and the observation, along with some interaction feedback or auxiliary information. The particular input format depends on the task domain and will be detailed in Subsection 4.1. The output part indicates the action decision. Specifically, we propose to present the action

```
Last 5 Actions:
search[3 ounce bottle bright citrus deodorant
  sensitive skin]
Observation:
Instruction:
i would like a 3 ounce bottle ...
[Back to Search]
...
[B078GWRC1J]
...
[B078GTKVXY]
...
[B08KBVJ4XN]
...
Available Actions:
back to search
...
```

```
Encouraged:
click[b078gwrc1j] -> 1.0 b078gwrc1j and
  b078gtkvxy are bright citrus deodorant less than
  50 dollars. I can check b078gwrc1j first.
Discouraged:
click[b087wksr2g] -> 0.0 b087wksr2g is not
  the desired item.
```

Figure 4: An exemplar for WebShop task set [Yao et al., 2022a]. The input part is depicted in the upper box and the output part is depicted in the lower box. Action candidates are advised along with their $Q$ value estimations and some optional extra information.

decisions in a form of "action advice" comprising both encouraged and discouraged actions rather than simply present an action to execute. This is motivated by the perspective that "*reasoning is*

*remembering*" to exploit both successful and failed experiences. To form the output part in the exemplar, the actions with the highest $Q$ value estimations from the retrieved record are given as the encouraged actions, while the actions with poor $Q$ value estimations (*e.g.*, zero or negative estimations) are given as the discouraged actions. It is believed that the advice with high value expectations can lead the LLM to follow the past success, while the advice with poor expectations will teach the LLM to avoid a similar failure. A clear depiction of the exemplar format can be found in Figure 4. Prompted by such exemplars, the LLM will also predict both encouraged and discouraged actions and speculate their $Q$ values given a new input. The predicted $Q$ values are used to select the optimal action, to be specific, the encouraged action with the highest $Q$ value speculation will be executed in the environment.

It is worth noting that REMEMBERER agent necessitates only a limited number of training steps to achieve a promising performance, which leads to a non-exhaustive action record within its memory. Consequently, instances may arise where there is only one action associated with a given context $(g, o_t)$, or the highest $Q$ value remains deficient, or no sufficiently unfavorable action exists to discourage. In such cases, randomly sampled action advice is favored over encouraging an action with low expectations or discouraging an action with moderate expectations. Our ablation study in Subsection 4.4 sheds light on various strategies for generating advice in such scenarios.

# 4 Experiments & results

## 4.1 Experiment setup & implementation details

To assess the effectiveness of REMEMBERER, we evaluate it on two recent task sets with the promising performance of LLM-based agents: WebShop and WikiHow. All the experiments are conducted based on the OpenAI API of GPT-3.5 [Brown et al., 2020] text-davinci-003[2].

**WebShop [Yao et al., 2022a]** WebShop is a task set simulating a web store site. The agent is instructed to browse the site and shop for the target goods. The information of over 1M products is crawled from the Amazon store[3]. About 12K product requests are re-written by crowd laborers to generate more diverse instructions. A score between 0 and 1 will be rated after shopping by assessing the correspondence between the product and the instruction. We followed Shinn et al. [2023] and conducted our experiments on the first 100 tasks from the same shuffled task list released along with the task set. At each interaction step, the LLM takes the web page representation and a list of available actions as input. The task instruction is omitted, for there is always an instruction present on the top of the web page. As there are no intermediate rewards during the episode, only the last 5 performed actions serve as procedure feedback. Inspired by the chain-of-thought technique [Wei et al., 2022] and the ReAct mechanism [Yao et al., 2022b], the LLM is prompted to predict a reason for its decision as the extra information depicted in Figure 4. The representation of the web pages is simplified in the same way as ReAct. The task similarity $f_g$ is calculated using the all-MiniLM-L12-v2 model from Sentence-Transformers [Reimers and Gurevych, 2019]. As it is noticed that the web pages in WebShop are instantiated from some templates, we categorize the web pages into four patterns and design a similarity lookup table to compute the observation similarity $f_o$ according to the web page patterns. The details about the similarity table should be referred to in the supplementary. It is observed that most of the tasks end in 5 steps, thus we directly conduct a full-trajectory expanding while performing multi-step bootstrapping:

$$Q'(o_t, a_t) = \sum_{\tau=t}^{T} \gamma^{\tau-t} r_\tau. \tag{7}$$

**WikiHow [Zhang et al., 2023]** WikiHow is a task set based on the collaborative wiki app WikiHow[4] running on the interaction platform Mobile-Env [Zhang et al., 2023]. The task set contains amounts of navigation tasks. The target of the agent is to follow the instructions and navigate to the required page. Intermediate rewards and instructions may be triggered during the episode. We followed Zhang et al. [2023] and evaluated the proposed REMEMBERER on the "canonical subset" comprising 70

---

[2] https://openai.com/api/
[3] https://www.amazon.com/
[4] https://www.wikihow.com/Main-Page

Table 1: Results on WebShop. The result of the prior state of the art, ReAct [Yao et al., 2022b], is attained with the public implementation released by the original authors. The RL, IL, and IL+RL results are retrieved directly from Yao et al. [2022a].

| Method | Avg Score | Success Rate |
|--------|-----------|--------------|
| LLM only | 0.55 | 0.29 |
| ReAct | 0.66 | 0.36 |
| RMMBR. | **0.68** | **0.39** |
| RL | 0.55 | 0.18 |
| IL | 0.60 | 0.29 |
| IL+RL | 0.62 | 0.29 |

Table 2: Results on WikiHow. "Mobile-Env" indicates the prior result from Zhang et al. [2023]. "RMMBR. (A)" denotes the results by directly running the evaluation of REMEMBERER with a human-annotated experience memory.

| Method | Avg Reward | Success Rate |
|--------|------------|--------------|
| LLM only | 2.58 | 0.90 |
| Mobile-Env | 2.50 | 0.89 |
| RMMBR. | **2.63** | **0.93** |
| RMMBR. (A) | 2.56 | 0.91 |

Table 3: Results on WebShop with different exemplar combinations (initial experiences for REMEMBERER) and different training sets (for REMEMBERER). $E_i$ denotes the different exemplar combinations, while $S_i$ denotes the different training sets. The first line of each method shows the mean scores, and the second line shows the success rates.

| | Different (Initial) Exemplars | | | Different Training Sets | | |
|--------|---------------|---------------|---------------|---------------|------|------|
| | $E_0 + S_0$ | $E_1 + S_0$ | $E_2 + S_0$ | $E_0 + S_1$ | Avg | Std |
| ReAct | **0.72** | 0.65 | 0.60 | - | 0.66 | 0.06 |
| | **0.42** | 0.35 | 0.30 | - | 0.36 | 0.06 |
| LLM only | 0.52 | 0.54 | 0.59 | - | 0.55 | 0.04 |
| | 0.26 | 0.28 | 0.32 | - | 0.29 | 0.03 |
| RMMBR. | 0.66 | **0.71** | **0.66** | **0.67** | **0.68** | **0.02** |
| | 0.37 | **0.41** | **0.37** | **0.40** | **0.39** | **0.02** |

tasks. Specifically, the LLM is input with the task description, the screen representation, and the step instruction. The screen is represented in an HTML element sequence following Zhang et al. [2023]. Additionally, the last 5 performed actions along with the last reward are given to the LLM as the procedure feedback. As for the output, the LLM is prompted to print the HTML representation of the operated element as the extra information. This is expected to force the LLM to discover the relation between the element id and the certain element. The task similarity $f_g$ designed for WikiHow is computed from the step instructions. It is noticed that the instructions follow some patterns, thus, we inspect the instructions and categorize them into six types. Then a similarity lookup table is designed according to the instruction types. The details should be referred to in the supplementary. The observation similarity $f_o$ is computed based on the length of the longest common sequence of the HTML elements in the screen representation:

$$f_o(sc_1, sc_2) = \frac{lcs(sc_1, sc_2)}{\max\{len(sc_1), len(sc_2)\}}. \tag{8}$$

The full-trajectory expanding is adopted, as most of the tasks will end in 5 steps as well.

## 4.2 Results on WebShop

REMEMBERER is applied to WebShop with 2-shot in-context learning. The experience memory is initialized with four annotated experiences of the decision step from one trajectory. The agent is trained for 3 epochs on a training set containing 10 different tasks outside the test sets used by Yao et al. [2022b] and Shinn et al. [2023]. To control the total expense and achieve bootstrapping, the succeeded tasks in the first epoch are excluded from training in the following two epochs. The trajectories exceeding 15 steps are considered to be failed, as most of the tasks can end in 5 steps. The main results are shown in Table 1. We used the public ReAct [Yao et al., 2022b] implementation released by the authors and run with text-davinci-003 instead of text-davince-002 in Yao et al. [2022b]. The run of ReAct shares the same trajectory as the exemplar with REMEMBERER. The "LLM only"

Table 4: Comparison of the number of annotated trajectories and steps of REMEMBERER and the IL baseline. The number of steps of the training set of IL is estimated according to the average human trajectory length on the test split as 11.3 in Yao et al. [2022a].

| Method | #Trajectories | #Steps |
|--------|---------------|--------|
| IL | 1,012 | ~11,436 |
| REMEMBERER | 1 | 4 |

Table 5: Comparison of the number of the tasks in the training set and the updating steps of RE-MEMBERER with the IL and RL baselines. The number of the updating steps of IL is estimated from 10 epochs on 1,012 trajectories with an average trajectory length of 11.3.

| Method | #Tasks | #Steps |
|--------|--------|--------|
| RL | 10,587 | 100,000 |
| IL | - | ~114,356 |
| REMEMBERER | 10 | 74 |

Table 6: Results on WikiHow with different exemplar combinations (initial experiences for REMEM-BERER) and different training sets (for REMEMBERER).

| | Different (Initial) Exemplars | | | Different Training Sets | | |
|---|---|---|---|---|---|---|
| | $E_0 + S_0$ | $E_1 + S_0$ | $E_2 + S_0$ | $E_0 + S_1$ | Avg | Std |
| LLM only | 2.56 | 2.60 | **2.59** | - | 2.58 | **0.02** |
| | 0.90 | 0.90 | 0.89 | - | 0.90 | **0.01** |
| RMMBR. | **2.63** | **2.63** | **2.59** | **2.66** | **2.63** | 0.03 |
| | **0.93** | **0.91** | **0.90** | **0.97** | **0.93** | 0.03 |

baseline indicates a single LLM with 2 fixed exemplars sampled from the initial experiences of REMEMBERER. The average performance of REMEMBERER exceeds the baseline by a large extent and surpasses the prior state of the art, ReAct, as well. This proves the effectiveness of augmenting the LLM with an external evolvable experience memory. The proposed REMEMBERER also outperforms the RL, IL (imitation learning), and IL+RL baselines on both metrics.

In order to verify the robustness of REMEMBERER, experiments with different initial experience combinations or a different training set are conducted. The results are depicted in Table 3. The initial experience combination $E_0$ denotes the certain trajectory adopted by the original implementation of ReAct while $E_1$ and $E_2$ are randomly sampled from $S_0$. It is observed that the proposed RE-MEMBERER can achieve better and more stable results with different initialization and training sets compared to ReAct. Thus, REMEMBERER can mitigate the workload to some extent to search for an optimal exemplar combination.

We compare the training efficiency of REMEMBERER with the conventional IL and RL methods in Table 4 and Table 5. In contrast to the IL, REMEMBERER requires quite few annotated samples to initialize the experience memory, while IL is in need of much more human annotations. REMEMBERER agent can be trained on only 10 tasks for 74 steps, while the RL and IL are expected to be trained for about 100 thousand steps to achieve an acceptable performance. Consequently, the proposed REMEMBERER offers a much more efficient way to build a practical agent agilely.

## 4.3 Results on WikiHow

REMEMBERER is applied to WikiHow with 2-shot in-context learning. The experience memory is initialized with two annotated experiences of the decision step. The agent is trained for 3 epochs on a training set containing 10 different tasks selected from WikiHow excluding the test tasks. Similar to the experiments on WebShop, the succeeded tasks in the first epoch are excluded from training in the following two epochs. As observed that most of the tasks require an interaction of less than 5 steps, the trajectory exceeding 15 steps will be regarded as failed. The main results are depicted in Table 2. The exemplars of the "LLM only" baseline are the initial experiences of REMEMBERER. The proposed REMEMBERER surpasses the baseline as well as the original result in Zhang et al. [2023]. In addition, 10 tasks are annotated to form an annotated experience memory. REMEMBERER agent with this annotated experience memory is evaluated without further training, and the result is denoted as "RMMBR. (A)" in the table. This result demonstrates that REMEMBERER is capable of

Table 7: Comparison of the average reward estimation of the full model and the ablation model without bootstrapping policy. The error is the absolute difference between the average reward estimation from the experience memory and the real training reward.

| Task Set | Setting | Avg Reward Estimation | Avg Training Reward | Abs Error |
|----------|---------|----------------------|---------------------|-----------|
| WebShop | Full Model | 0.86 | 0.84 | **0.02** |
|  | w/o bootstrp. | 0.62 | 0.84 | 0.22 |
| WikiHow | Full Model | 2.48 | 2.60 | **0.12** |
|  | w/o bootstrp. | 1.98 | 2.70 | 0.72 |

Table 8: Results of ablation study

| Task Set | Setting | Avg Reward/Score | Success Rate |
|----------|---------|------------------|--------------|
| WebShop | Full Model | 0.66 | 0.37 |
|  | w/o bootstrp. | 0.67 | 0.36 |
|  | w/o random | 0.65 | 0.37 |
| WikiHow | Full Model | 2.63 | 0.93 |
|  | w/o bootstrp. | 2.54 | 0.89 |
|  | w/o random | 2.64 | 0.90 |
|  | w/o discouraged | 2.48 | 0.81 |
|  | w/o task sim. $f_g$ | 2.63 | 0.94 |
|  | w/o obsrv. sim. $f_o$ | 2.47 | 0.87 |

exploiting expert experiences, which can be regarded as analogous to conventional imitation learning. Nevertheless, the annotated experiences may not offset the exact shortage of the particular LLM. In contrast, the RL training will have an opportunity to collect more specific experiences and gain a more promising performance.

The experiments with different initial experience combinations or a different training set are conducted on WikiHow as well, and the results are shown in Table 6. The proposed REMEMBERER achieves a consistent improvement compared to the baseline with fixed exemplars, which proves the effectiveness and robustness of REMEMBERER.

## 4.4 Ablation study

Several ablation studies are conducted to verify the design of REMEMBERER framework.

**Ablation on $n$-step bootstrapping policy** Ablation studies are conducted to verify the necessity of $n$-step bootstrapping policy to update the $Q$ value estimations in the experience memory. As stated in Subsection 3.2, updating without bootstrapping may lead to inaccurate value estimations owing to few training steps to explore and exploit. In order to verify this perspective, an average reward estimation is calculated by averaging the sum of the maximum $Q$ value and the history reward stored for each observation in the experience memory:

$$\hat{R} = \frac{1}{M} \sum_{i=1}^{M} (R_h(g_i, o_i) + \max_a Q(g_i, o_i, a)), \tag{9}$$

where $R_h$ denotes the total reward of the steps before $(g_i, o_i)$ on the trajectory and $M$ is the size of the memory. The deduced average reward estimation $\hat{R}$ is compared to the real training reward, and an absolute error is calculated in Table 7. It can be observed that the reward estimation from the experience memory trained without bootstrapping suffers a far greater error than that with bootstrapping. Meanwhile, the performance on the test set is demonstrated in Table 8. Although there is no apparent disparity in the final performance on the WebShop task set, a visible degradation is observed on WikiHow, which reveals the latent risk of a non-bootstrapping update.

**Ablation on the advice generation strategy**   As stated in Subsection 3.3, owing to the non-exhaustive exploration in the brief training stage, there may be no suitable candidates for the action advice in the exemplars. For instance, there may be no actions recorded with a poor enough $Q$ value estimation or no actions recorded as high-reward. Under this case, action advice can be generated with a randomly sampled action that is not in the record, or it can be given by directly encouraging the action with the highest $Q$ value estimation and discouraging the action with the lowest estimation without regard to the certain value. These two strategies are compared in Table 8. As the results illustrate, the random plan appears to have a minor superiority over the non-random plan. This is attributed to that advice with improper value expectations will mislead the LLM to take wrong judgments about the true value of the available actions.

Additional experiments are conducted to investigate the necessity of the discourage actions in the output part of exemplars and the impact of similarity function components. Owing to limit of budgets, these experiments are only conducted on WikiHow task set.

**Ablation on necessity of the discouraged actions**   The proposed output format "action advice" comprises both encouraged and discouraged actions. The discouraged actions are believed to help the LLM to avoid similar failures. Results in Table 8 prove necessity of the discouraged actions. Without access to the discouraged actions, the agent can only achieve a much poorer performance than the full model. In the case shown in the supplementary, it can be seen that there may not be proper actions to encourage in the retrieved experience. In such cases, the discouraged actions are especially crucial for the agent to prevent repeating similar mistakes.

**Ablation on the similarity function**   As stated in Subsection 3.3, a similarity function is required to select related experiences from the memory. In experiments, the similarity is implemented as two components: task similarity $f_g$ and observation similarity $f_o$. Ablation studies are conducted to draw a brief perspective on the impact of these two components. As shown in Table 8, removal of task similarity seems not to affect the performance remarkably, while removal of observation similarity causes a serious degradation. This may indicate that on these tasks, the tested LLM benefits more from experiences that have similar observations rather than similar instruction patterns. On the other side, the pattern-based task similarity for WikiHow introduced in Subsection 4.1 may be too coarse to cluster the experiences. During interaction, the agent may receive instructions of the same pattern (*e.g.*, "access article ABC") while facing different types of observation (*e.g.*, search result page or category page). The appropriate actions in two situations are also different. Removal of observation similarity will eliminate this difference in experience selection and results in misleading. Case study in the supplementary shows this perspective.

## 5   Conclusion

We introduce Reinforcement Learning with Experience Memory (RLEM) to aid the LLM in learning from its interaction experiences for decision-making tasks. A novel LLM-based agent framework called REMEMBERER is then designed with RLEM by equipping the LLM with a persistent experience memory and updating the memory with the RL algorithm. REMEMBERER agent is capable of exploiting the interaction experiences to improve its policy and gains a significant improvement compared to the baseline. Our experimental results demonstrate the superiority. Owing to the simplicity and effectiveness of REMEMBERER, we believe that this work provides a valuable perspective on designing evolvable LLM-based agents with RLEM.

## 6   Limitations

The proposed REMEMBERER agent demonstrates strong superiority on the tested benchmarks. Nevertheless, it is wondered how this framework will be applied to the environments with more long-term episodes or with more extensive or visual-rich observations. Besides, it is observed that the performance of REMEMBERER will encounter quick saturation in training process. This may be due to the limited number of active exemplars. Further efforts are expected to be dedicated in to make the agent performance evolve continuously. Furthermore, as an early exploration, we didn't make use of complicated RL techniques. How recent advancement in RL domain works under RLEM is also an interesting problem.

## Acknowledgements

This work is funded by the China NSFC Project (No.62106142 and No.62120106006) and Shanghai Municipal Science and Technology Major Project (2021SHZDZX0102).

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
