# 1 Details about the observation formats

---

Instruction:
i would like a 3 ounce bottle of bright citrus deodorant for sensitive skin, and price lower than 40.00 dollars
[Back to Search]
Page 1 (Total results: 50)
[Next >]
[B078GWRC1J]
Bright Citrus Deodorant by Earth Mama | Natural and Safe for Sensitive Skin, Pregnancy and Breastfeeding, Contains Organic Calendula 3-Ounce
$10.99
[B078GTKVXY]
Ginger Fresh Deodorant by Earth Mama | Natural and Safe for Sensitive Skin, Pregnancy and Breastfeeding, Contains Organic Calendula 3-Ounce
$10.99
[B08KBVJ4XN]
Barrel and Oak - Aluminum-Free Deodorant, Deodorant for Men, Essential Oil-Based Scent, 24-Hour Odor Protection, Cedar & Patchouli Blend, Gentle on Sensitive Skin (Mountain Sage, 2.7 oz, 2-Pack)
$15.95

---

Figure 1: Example of the observation of WebShop

The observation of WebShop is simplified based on the `text_rich` format of WebShop [Yao et al., 2022a] in exactly the same way with Yao et al. [2022b]. Specifically, the HTML markups are omitted, and the buttons are represented in `[text]` or `[[text]]` instead of the complicated `[button] text [button_]` or `[clicked button] text [clicked button_]`. Furthermore, the number of displayed search results per page is clipped to 3 instead of 10. An example is shown in Figure 1.

The observation of WikiHow is represented in exactly the same way with Zhang et al. [2023]. Specifically, the page is converted into a sequence of HTML elements corresponding to the visible leaf nodes on the Android ™ view hierarchy (VH). The node classes are converted into HTML tags and a few VH properties are converted into similar HTML attributes. The `text` property is converted to the text content of the common HTML element or the `value` attribute of the `input` element.

## 2 Lookup table of the pattern-based similarity functions

### 2.1 Lookup table of the page similarity function of WebShop

We inspected the pages from WebShop and categorized them into 4 patterns as depicted in Table 1.

Table 1: Patterns of WebShop pages

| Pattern | Description |
| --- | --- |
| search | The page to search for an item |
| itemlisting | The page listing the search results |
| item | The information page of a specific item |
| others | The item description page, item feature page, and review page |

The similarity lookup table is defined in Table 2.

Table 2: Lookup table of the page similarity of WebShop

|  | search | itemlisting | item | others |
|---|---|---|---|---|
| search | 1 | 0 | 0 | 0 |
| itemlisting | 0 | 1 | 0 | 0 |
| item | 0 | 0 | 1 | 0.3 |
| others | 0 | 0 | 0.3 | 1 |

## 2.2 Lookup table of the instruction similarity function of WikiHow

We inspected the step instructions from WikiHow and categorized them into 6 patterns as depicted in Table 3.

Table 3: Patterns of WikiHow instructions

| Pattern Name | Pattern Template |
|---|---|
| search | Search an article to learn ... |
| article | Access the article ... |
| author | Check the author page of ... |
| category | Access the page of category ... |
| reference | Check the reference list. |
| about | Access the about page ... |

Table 4: Lookup table of the instruction similarity of WikiHow

|  | search | article | author | category | reference | about |
|---|---|---|---|---|---|---|
| search | 1 | 0.1 | 0 | 0 | 0 | 0 |
| article | 0.1 | 1 | 0.3 | 0.3 | 0 | 0 |
| author | 0 | 0.3 | 1 | 0.8 | 0.3 | 0.3 |
| category | 0 | 0.3 | 0.8 | 1 | 0.3 | 0.3 |
| reference | 0 | 0 | 0.3 | 0.3 | 1 | 0.8 |
| about | 0 | 0 | 0.3 | 0.3 | 0.8 | 1 |

The similarity lookup table is defined in Table 4.

## 3 Hyper-parameters

The discount factor $\gamma$ to accumulate the rewards in the formula of $Q$ value is 1, which means no discounts are considered. The learning rate $\alpha$ is $1/N$ where $N$ denotes the times the value is updated. Such a learning rate is chosen, as the tested environments are stationary and each estimation to the value is expected to be equally weighted. The similarity weight factor $\lambda$ is 0.5, indicating two parts of the similarity function contribute equally.

## 4 Capability evolving of REMEMBERER

We further conducted experiments to see how the capability of REMEMBERER evolves during training. Owing to the limit of budgets, a subset of only 20 tasks is sampled from the full test set. We visualize the performance on the subset of REMEMBERER at epochs 1, 5, and 10. The performance at epoch 3, which is used for the experiments in the main paper, is visualized as well. The visualization is available in Figure 2. It can be seen that the performance of REMEMBERER improves during the training procedure. However, there seems to be a saturation for the performance, which may be attributed to the limited number of the active exemplars and training tasks. The saturation of the average reward comes later than that of the success rate. This fact indicates that REMEMBERER

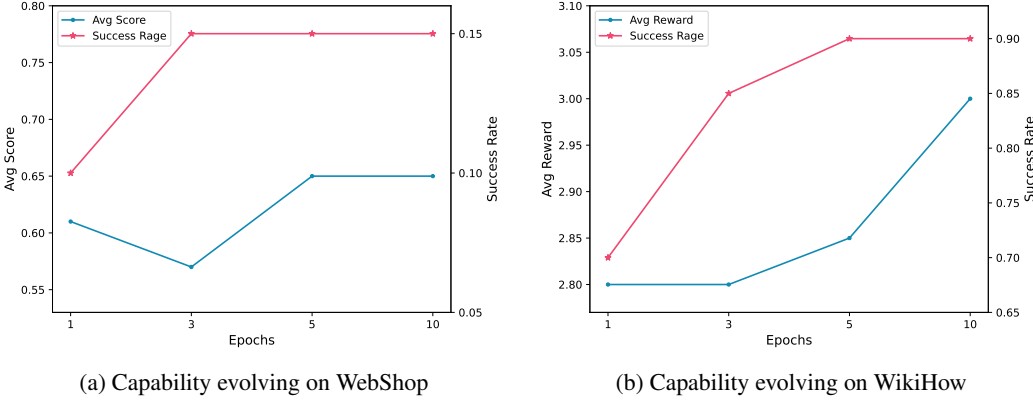

(a) Capability evolving on WebShop

(b) Capability evolving on WikiHow

Figure 2: Performance on a random subset at epochs 1, 3, 5, and 10

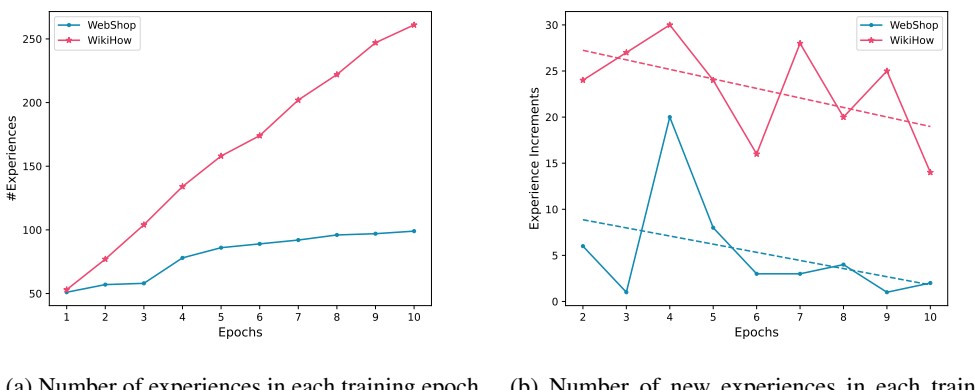

(a) Number of experiences in each training epoch

(b) Number of new experiences in each training epoch. The dashed lines are acquired by performing the least squares fit to the data points.

Figure 3: Variation of the experience number in the training process

can still seize more rewards through training on several unsuccessful tasks even the success rate has already saturated. In other words, the hard tasks benefit more from the later phase of training than the easy tasks. Besides, REMEMBERER reaches saturation on WebShop earlier than on WikiHow. To give an explanation, the number of the experiences in the memory after each training epoch is inspected. As shown in Figure 3, there are much fewer new experiences added into the memory in the later epochs for WebShop than for WikiHow. The certain reason may be due to the specific training set or some internal characteristics of the task domain, which will be further investigated in the future work.

## 5 $Q$ function fitting ability of REMEMBERER

Ablation study in the main paper has demonstrated that $n$-step bootstrapping manages to improve precision of the learned $Q$ values in the memory. This section will give further discussion about over-estimation of learned $Q$ values in the memory and whether the LLM can learn the certain $Q$ function through in-context learning (ICL).

Double Q-Learning [van Hasselt, 2010] is usually leveraged to ameliorate over-estimation for lookup-based Q-Learning. Table 5 shows the $Q$ value estimation results with Double Q-Learning applied. Over-estimation does be suppressed, however, serious under-estimation is introduced, and the estimation error fails to ameliorate. This is explained by that Double Q-Learning iteratively

Table 5: Comparison of the average reward estimation of the full model and the Double Q-Learning model

| Task Set | Setting | #Epochs | Avg Reward Estimation | Avg Training Reward | Abs Error | Relative Error |
|---|---|---|---|---|---|---|
| WebShop | Full Model | 3 | 0.86 | 0.84 | **0.02** | **2.38** |
| | +DoubleQL | 3 | 0.71 | 0.75 | 0.04 | 5.33 |
| | +DoubleQL | 6 | 0.69 | 0.77 | 0.08 | 10.39 |
| WikiHow | Full Model | 3 | 2.48 | 2.60 | **0.12** | **4.62** |
| | +DoubleQL | 3 | 2.47 | 2.90 | 0.43 | 14.83 |
| | +DoubleQL | 6 | 2.70 | 2.90 | 0.20 | 6.90 |

updates two $Q$ value lookups and requires more steps to converge to an accurate enough estimation. In contrast, plain Q-Learning performs better in few-step circumstances.

As regards whether the LLM learns the certain $Q$ value function, predicted values of LLM during the test phase on WebShop are inspected. The average absolute error is 0.417. This fact indicates that the LLM does not really learn the certain $Q$ function, as the reward in WebShop is always between 0 and 1. Nevertheless, the LLM can still predict the appropriate actions. This is due to the inessentiality of absolutely precise $Q$ value prediction during test. It is the relative relation between the values of candidate actions that is truly important. Once LLM can distinguish the valuable actions from candidates, it can take the right policy.

## 6 Example of the exemplars

An example of the input exemplar for WebShop and WikiHow is given in Figure 4 and Figure 5, respectively.

## 7 Case study

Figure 6 gives a case from the ablation study on necessity of the discouraged actions. If the discouraged actions are omitted in the action advice from an experience without encouraged actions, the LLM will have no ability to avoid failures of the same pattern.

A case from the ablation study on the similarity function on WikiHow task set is depicted in Figure 7. Once the observation similarity $f_o$ is omitted, the agent will retrieve experience only according to the instruction and cannot adjust the selection in accordance with the particular observation. This will cause improper experience retrieval and lead to poorer performance.

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

```
---
Last 5 Actions:
- search[3 ounce bottle bright citrus deodorant sensitive skin]
Observation: |

  Instruction:
  i would like a 3 ounce bottle of bright citrus deodorant for sensitive skin, and
  price lower than 40.00 dollars
  [Back to Search]
  Page 1 (Total results: 50)
  [Next >]
  [B078GWRC1J]
  Bright Citrus Deodorant by Earth Mama | Natural and Safe for Sensitive Skin,
  Pregnancy and Breastfeeding, Contains Organic Calendula 3-Ounce
  $10.99
  [B078GTKVXY]
  Ginger Fresh Deodorant by Earth Mama | Natural and Safe for Sensitive Skin,
  Pregnancy and Breastfeeding, Contains Organic Calendula 3-Ounce
  $10.99
  [B08KBVJ4XN]
  Barrel and Oak - Aluminum-Free Deodorant, Deodorant for Men, Essential
  Oil-Based Scent, 24-Hour Odor Protection, Cedar & Patchouli Blend, Gentle
  on Sensitive Skin (Mountain Sage, 2.7 oz, 2-Pack)
  $15.95
Available Actions:
- back to search
- next >
- b078gwrc1j
- b078gtkvxy
- b08kbvj4xn
...

Encouraged:
click[b078gwrc1j] -> 1.0 b078gwrc1j and b078gtkvxy are bright citrus
deodorant less then 50 dollars. I can check b078gwrc1j first.
Discouraged:
click[b087wksr2g] -> 0.0 b087wksr2g is not the desired item.
```

Figure 4: Exemplar for WebShop. YAML markups are adopted to avoid confusing the keywords like "Observation:" with the colon-ended titles in the page representation.

Danyang Zhang, Lu Chen, and Kai Yu. Mobile-Env: A universal platform for training and evaluation of mobile interaction. *CoRR*, abs/2305.08144, 2023. URL https://arxiv.org/abs/2305.08144.

```
Task:
Search an article to learn how to hide gauges.
Then, access the article "How to Hide Gauges"
Last 5 Actions:

Screen:
<button alt="Open navigation drawer" id="0" clickable="true"></button>

<div class="webView" id="3" clickable="true"></div>
<div class="statusBarBackground" id="4" clickable="false"></div>
Instruction:

Last Reward:
0.0
Total Reward:
0.0
---

Encouraged:
INPUT(2, hide gauges) -> 2.0 <img class="search button" alt="Search" id="2"
clickable="true">
Discouraged:
SCROLL(RIGHT) -> 0.0
```

Figure 5: Exemplar for WikiHow

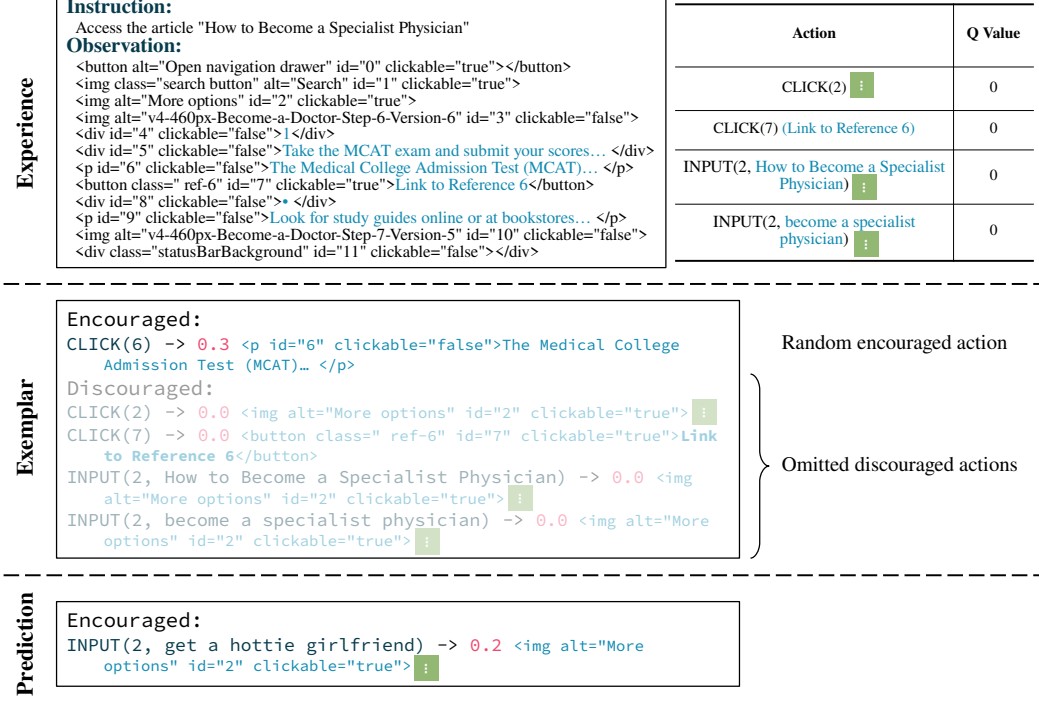

Figure 6: Case of the ablation study on the discouraged actions. As there are no valuable actions to encourage in the experience, a random action is generated. When the discouraged actions with low value are omitted, the LLM may repeat the failure with the same pattern.

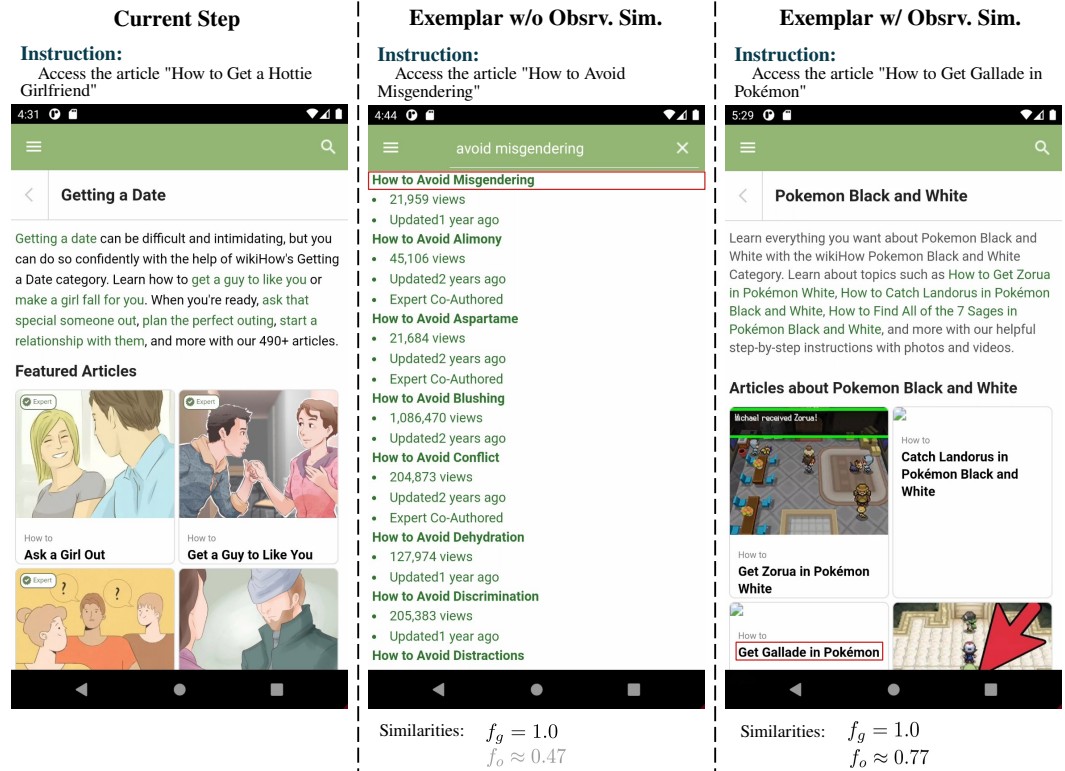

Figure 7: Case of the ablation study on the similarity function. Encouraged actions recorded in the experiences are marked by red rectangles.