# OpenReview forum: "Large Language Models Are Semi-Parametric Reinforcement Learning Agents"
_NeurIPS.cc/2023/Conference — NeurIPS 2023 poster_

### Official Review · Reviewer_d9XL · 2023-07-03

**Soundness:** 3 good
**Presentation:** 2 fair
**Contribution:** 3 good
**Rating:** 5
**Confidence:** 4

**Summary:**

The paper proposes an LLM + RL architecture for text-based domains. The key idea is to run Q-learning on the side and augment LLM's prompt with Q values for available actions. The evaluation is done one WikiHow and WebShop.

Acknowledging the rebuttal, I'm satisfied with authors responses and happy to increase the score.

**Strengths:**

The paper's core idea---adding q-values to LLM prompts---is interesting.

**Weaknesses:**

Clarity is the main issue with the paper. It is hard to read and hard to understand. Here are some of the issues that stand out:
- In section 3.2 the Q-learning is introduced. It is not clear how exactly it is performed. Is function approximation used? If so then what is the network architecture. Is it tabular? If so then how come we are encountering exactly the same observations in the test set?
- On line 163 the term "flattening" is used. This is not a common term in the literature, what is meant by it exactly? the eq.4 shgiws  bootstrapping for Q-value estimation. Is that what is meant?
- There is a substantial amount of heuristics that is used to make the system work, but they are not well explained and their influence on the results is not well evaluated. For example, line 227-232 describes categorisation of the web pages into categories. Is this used for q learning? What happens if this categorisation is not used? Same with 243-249 for wikihow.
- It is not clear whether the difference to the baselines is significant. For example on WebShop ReAct has avg score of 66.6 (from original paper) and the proposed method has 68. Is that a significant difference? Success rate is 40 for ReAct in the original paper, but is reported 36 in this paper. Where does the difference come from? Since the proposed method achieves the success rate of 38 this warrants a clarification.
- Positioning in the introduction is adding to the confusion. The conversation of about using the external memory is misleading, as what is later proposed is Q-learning augmentation.

**Questions:**

Would authors please address the points outlined in the weaknesses section.

**Limitations:**

The paper does not have limitations section

---

> ### Author Rebuttal · Authors · 2023-08-07
>
> Thanks for your kind review.
>
> **About Q-Learning and $Q$ function architecture**: The $Q$ function is
> implemented with the experience memory as a lookup table. Q-Learning is applied
> to the experience-memory-based lookup. Note that, we don't use this lookup to
> predict a $Q$ value for the new observation during inference. All the records
> in the lookup come from the training set and serve as candidate exemplars in
> the input prompt to LLM. During the test, it is the LLM that will speculate a $Q$
> value for new observations and distinguish good actions from the bad ones. Here
> is a naive example:
>
> Experience Memory:
>
> | Task  | Observ. | Act.  | $Q$   |
> |-------|---------|-------|-------|
> | $g_0$ | $o_0$   | $a_0$ | $q_0$ |
> | $g_1$ | $o_1$   | $a_1$ | $q_1$ |
>
> Assume the new observation (& task) encountered during test is $(g, o)$. And
> several similar experiences will be retrieved from the memory, *e.g.*, $(g_0,
> o_0)$ and $(g_1, o_1)$. Then the LLM will be fed:
>
> $$
> g_0, o_0: a_0 \rightarrow q_0
> $$
> $$
> g_1, o_1: a_1 \rightarrow q_1
> $$
> $$
> g, o:
> $$
>
> The LLM is prompted to predict actions in format $a \rightarrow q$ for $(g, o)$.
>
> **About "flattening"**: Eqn. 4 does show what is "flattening", which means we
> flatten/expand the preceding $n$ steps in the Bellman equation. We find that
> this is usually called $n$-step Q-Learning or $n$-step bootstrapping [1, 2] and
> we will update the expression in later revision accordingly.
>
> **About categorization in Sec. 4.1**: The categorization of pages in WebShop
> and instructions in WikiHow is used to calculate the similarity function
> introduced in Sec. 3.3. The similarity function itself is not related with
> Q-Learning procedure, but is used to select relative exemplars in the input
> prompt to LLM for in-context learning. We will improve the corresponding
> explanation and make it more clear in later revision.
>
> **About ReAct results**: The reported ReAct results in this paper are obtained
> with *GPT LLM text-davinci-003* on 100 test tasks, while the results reported
> in the original paper of ReAct are obtained with *PaLM LLM* on 500 test tasks.
> That's why the success rate reported in our Tab. 3 is **44**, while the success
> rate in the original paper is 40. The success rate 36 in Tab. 1 is an
> **average** result across three different exemplars (the result 44 takes exact
> the same exemplar with the original paper of ReAct). You can also refer to the
> average result 36 in "Avg" column in Tab. 3. Instead of significance of
> improvement, we mainly claim the superior *robustness to the initial exemplars*
> compared to ReAct, as shown by the results across different exemplars in Tab.
> 3. It may take additional human labors and budget costs to find an optimal
> exemplar to guarantee the performance of the raw ReAct. In contrast, Rememberer
> holds a more stable performance across different initialization and can reduce
> the required human labors to construct appropriate exemplars.
>
> **Relation between Q-Learning and experience memory**: For this question, we refer you to global reply 1. We use RL to assist
> LLM in exploiting the interaction experience, and the experience memory serves
> as *pivot* between RL algorithm and LLM. In practice, the experience memory is
> regarded as a tabular $Q$ function and is updated by Q-Learning.
>
> **About limitations**: Thanks for your kind reminder. We will add the section
> of Limitations in later revision.
>
> * [1] Watkins, Christopher John Cornish Hellaby. Learning from delayed rewards.
>   PhD thesis, University of Cambridge, England, 1989.
> * [2] Peng, Jing and Williams, Ronald J. Incremental multi-step q-learning.
>   Machine Learning 1996.

---

> > ### Author Response · Authors · 2023-08-20
> > **Looking forward to your reply**
> >
> > Hello. The author-reviewer discussion period is going to end. We wonder if our rebuttal solves your concerns. If there remains questions, we are willing to conduct further and deeper discussion with you.

---

### Official Review · Reviewer_4pvM · 2023-07-04

**Soundness:** 3 good
**Presentation:** 3 good
**Contribution:** 3 good
**Rating:** 4
**Confidence:** 4

**Summary:**

The authors introduced Reinforcement Learning with Experience Memory (RLEM) to update the memory of the LLM agent, enabling it to evolve its capability without fine-tuning the parameters of the LLM. Extensive experiments were conducted on two RL task sets to evaluate the proposed framework. The experimental results demonstrate the superiority and robustness of Rememberer.

**Strengths:**

1. The interesting perspective of exploration is a meaningful attempt to combine LLM with RL.
2. The paper is well-written, easy to understand, and the illustrations are exquisite.
3. The ample experiments demonstrate the effectiveness of the proposed method.

**Weaknesses:**

The discussion and introduction about whether to use RL are not sufficient. It is unclear why RL must be used instead of other methods.

**Questions:**

1. Why use RL instead of other methods?
2. If there are alternative methods to RL, should they be supplemented with comparisons in the experimental section?

**Limitations:**

Refer to weaknesses and questions

---

> ### Author Rebuttal · Authors · 2023-08-07
>
> Thanks for your kind concern. First, we want to further clarify the necessity
> of updating the $Q$ values in the experience memory. Compared to the
> traditional ICL (in-context learning) methods with labeled dynamic exemplars,
> there will be both good and bad experiences in the memory of Rememberer. Hence,
> *it is necessary to discriminate between good and bad experiences through the
> learned $Q$ values*. Given this premise, reinforcement learning is the most
> straightforward method to update the recorded $Q$ values in POMDP
> (partially-observable Markov decision process) problems. And we didn't work out
> other alternatives.
>
> If your concern is about other LLM-based agents utilizing memory or experience,
> we are glad to have a quick comparison for you (Please note that most of these
> methods are released after the submission deadline of NeurIPS, thus, we were unable
> to include the comparison into our paper). Voyager [1] proposes to use codes as
> actions, and designs a skill library to store the successful programs as
> skills.  GITM [2] stores the past successful action sequences in a text memory
> to assist an LLM planner in future planning without further discriminating
> their values.  Then the stored experiences are summarized by LLM to gain deeper
> insights into planning policy. ChatDB [3] leverages relational database to
> track states in a dynamic process. In contrast, we store the experiences in a
> structured memory and learns their $Q$ values to filters out the more valuable
> actions. In this way, we can combine RL with LLM and design a semi-parametric
> RL agent. We are glad to include these comparisons in Related Work in later
> revision.
>
> As it will take some certain efforts to migrate these methods to the test
> benches in this paper, we may not introduce further new results during
> rebuttal.  Instead, we can draw a number of valuable inspirations from them and
> plan to include several new insights in our future work.
>
> * [1] Guanzhi Wang, Yuqi Xie, Yunfan Jiang, Ajay Mandlekar, Chaowei Xiao, Yuke
>   Zhu, Linxi Fan, Anima Anandkumar. Voyager: An Open-Ended Embodied Agent with
>   Large Language Models. arXiv:2305.16291.
> * [2] Xizhou Zhu, Yuntao Chen, Hao Tian, Chenxin Tao, Weijie Su, Chenyu Yang,
>   Gao Huang, Bin Li, Lewei Lu, Xiaogang Wang, Yu Qiao, Zhaoxiang Zhang, Jifeng
>   Dai. Ghost in the Minecraft: Generally Capable Agents for Open-World
>   Environments via Large Language Models with Text-based Knowledge and Memory.
>   arXiv:2305.17144.
> * [3] Chenxu Hu, Jie Fu, Chenzhuang Du, Simian Luo, Junbo Zhao, Hang Zhao.
>   ChatDB: Augmenting LLMs with Databases as Their Symbolic Memory.
>   arXiv:2306.03901.

---

> > ### Comment · Reviewer_4pvM · 2023-08-15
> > **Keep Score**
> >
> > Thanks for your reply, but the current reply still doesn't address my concerns, so I'm currently choosing to keep the score.

---

> > > ### Author Response · Authors · 2023-08-15
> > >
> > > Hello. Thanks for your reply. It is a pity that we didn't address your concerns. Maybe you can have further explanations for your questions, so that we can give a more specific reply.

---

> > > > ### Comment · Reviewer_4pvM · 2023-08-22
> > > > **Reviewers' responses**
> > > >
> > > > At the moment I still think that some RL (or other angle) comparison method is lacking, so I reserve my concerns

---

> > > > > ### Author Response · Authors · 2023-08-22
> > > > >
> > > > > It seems that we don't really understand your concerns. We wonder what the so-called "alternative methods" refer to? Method of **what** are we talking about? Is it the method to update the experience memory, or the method that we augment LLM with an external memory? Could you please give some examples of the so-called "alternative methods", *i.e.*, what they are, or for what they are ? If you could give some concrete clarification, we may handle your concerns better.

---

### Official Review · Reviewer_dPzh · 2023-07-06

**Soundness:** 2 fair
**Presentation:** 3 good
**Contribution:** 3 good
**Rating:** 4
**Confidence:** 3

**Summary:**

This paper proposes a framework to combine RL w/ LLM using an offline Q-learning setting. An experience memory component is proposed to store past experience for estimating Q values. Evaluation on WikiHow and WebShop demonstrate the effectiveness of the proposed method and framework.

**Strengths:**

The paper proposes an interesting idea of combing LLM w/ RL. This is a relatively new field. The evaluation results on 2 tasks are strong.

**Weaknesses:**

This work seems very empirical. One question I have is how to generalize this method to other tasks. It will be good to see some theoretical analysis.

**Questions:**

1. It might be better to not use OpenAI logo to represent LLM in the figures.
2. In Equation (1), how is max calculated?
3. It is not clear how the reward is defined and can be obtained.
4. There are a lot of RL issues people have overcome over years, such as over estimation of Q values, sparse reward, using replay buffer to provide iid data. It seems the proposed method will have all these issues. How did this paper tackle them? How do we guarantee that this method will converge?

**Limitations:**

It seems the proposed method of storing memory is not very scalable.

---

> ### Author Rebuttal · Authors · 2023-08-07
>
> Thanks for your valuable review and advice.
>
> **Generalizability**: For this question, we refer you to global reply 2.
>
> **$\max$ in Eqn. 1**: This $\max$ is calculated from the actions already
> recorded for $(g, o_{t+1})$, as we cannot traverse all the possible actions
> when there are free-form languages. We will clarify this in later revision.
> Actually, in the two tested task sets, the $Q$ value of the unrecorded actions
> can be deemed to have a default value 0. Such a "default" value may have other
> better choices in different task domains.
>
> **How reward is defined and obtained**: The reward is defined by the underlying
> task sets and obtained from the environments during interaction. As for
> WebShop, a score will be computed at the end as the returned reward according
> to the relevance between the reference product and the agent-bought product. As
> regards WikiHow, the agent is instructed to navigate several pages and a reward
> will be given when the agent manages to reach one of the target pages.  The
> details about the reward definition should be referred to in the original paper
> of the task sets.
>
> **Classic RL issues**: This paper mainly focuses on how to improve and evolve
> LLM's capability with RL algorithm, but not on the issues of RL itself.
> Therefore, we only implement a basic Q-Learning method, ignoring a few issues
> in advanced RL domains. The methods aiming at solving these RL issues (*e.g.*,
> Double Q-Learning [1], replay buffer [2], and diverse exploration methods) are
> orthogonal with this work, and will be straightforward to be embedded into
> Rememberer. Besides, there are a few statements w.r.t. the specific concerns.
>
> * Over-estimation. As shown in Tab. 7 in the paper, over-estimation doesn't
>   constitute a serious problem for the two tested task sets. Double Q-Learning
>   is usually adopted for lookup-based $Q$ estimators to ameliorate
>   over-estimation by iteratively updating two $Q$ estimators. However, such an
>   iterative updating method may require more update steps to ensure an accurate
>   enough estimation than current Rememberer. If time permits, we will try to
>   supplement several results obtained with Double Q-Learning.
> * Sparse reward. On the two tested task sets, the strong base capability of LLM
>   guarantees trajectories that won't deviate far from the optimal policy. Thus,
>   sparsity of reward will not constitute a serious issue.
> * Convergence. We cannot guarantee mathematical convergence of Q-Learning on
>   the experience memory, as it is difficult for lots of seriously bad
>   observations and actions to be accessed owing to the strong base capability
>   of LLM, *i.e.*, insufficient traversal of observation & action spaces.
>   However, the experience memory is trained based on the strong capability of
>   LLM. Therefore, the training process shouldn't deviate far from the optimum.
>   Or in other word, the seriously bad states have already been implicitly
>   traversed in the pretraining stage of LLM, thus, convergence shouldn't be a
>   crucial issue. On the other hand, the experience memory serves as an
>   augmentation to a capable foundation model, thus, the mathematical
>   convergence is not as critical as for an RL model from scratch.  The current
>   version can already help the LLM well.
>
> **Scalability of the method to store experiences**: The powerful basic
> capability of LLM makes a large-scale experience memory unnecessary. Meanwhile,
> the limited number of exemplars in the prompt will lead to a performance
> saturation when the size of experience memory increases continuously. We refer
> you to the experiments in Sec. 4 in the supplementary for a deeper insight into
> this argument. Besides, if there is truly a need for large-scale experience
> memory, more scalable methods to store and retrieve the experiences (*e.g.*,
> vector database implemented with FAISS) can be surely adopted. The concrete
> implementation of experience memory is not coupled with Rememberer framework.
>
> **About the figure**: Thanks for your kind reminder. We will update our figure
> in later revision.
>
> * [1] Hado van Hasselt. Double Q-learning. NeurIPS 2010.
> * [2] Volodymyr Mnih, Koray Kavukcuoglu, David Silver, Andrei A. Rusu, Joel
>   Veness, Marc G. Bellemare, Alex Graves, Martin A. Riedmiller, Andreas
>   Fidjeland, Georg Ostrovski, Stig Petersen, Charles Beattie, Amir Sadik,
>   Ioannis Antonoglou, Helen King, Dharshan Kumaran, Daan Wierstra, Shane Legg,
>   Demis Hassabis. Human-level control through deep reinforcement learning.
>   Nature 2015.

---

> > ### Author Response · Authors · 2023-08-13
> > **Supplementary result about over-estimation**
> >
> > Hello, we implemented Double Q-Learning and would like to supplement several
> > results here.
> >
> > **Estimation Error on WebShop**
> >
> > | Setting    | Epoch | Estimation | Real Training Reward | Abs Error | Relative Error |
> > |:-----------|:-----:|:----------:|:--------------------:|:---------:|:--------------:|
> > | Full Model |   3   |    0.86    |         0.84         |    0.02   |      2.38      |
> > | +DoubleQL  |   3   |    0.71    |         0.75         |    0.04   |      5.33      |
> > | +DoubleQL  |   6   |    0.69    |         0.77         |    0.08   |      10.39     |
> >
> > **Estimation Error on WikiHow**
> >
> > | Setting    | Epoch | Estimation | Real Training Reward | Abs Error | Relative Error |
> > |:-----------|:-----:|:----------:|:--------------------:|:---------:|:--------------:|
> > | Full Model |   3   |    2.48    |         2.60         |    0.12   |      4.62      |
> > | +DoubleQL  |   3   |    2.47    |         2.90         |    0.43   |      14.83     |
> > | +DoubleQL  |   6   |    2.70    |         2.90         |    0.20   |      6.90      |
> >
> > As shown by the results, iterative update of Double Q-Learning fails to
> > ameliorate the estimation error in *few-step* training. Over-estimation is
> > truly suppressed, however, serious under-estimation may be introduced. [1]
> >
> > * [1] Hado van Hasselt. Double Q-learning. NeurIPS 2010.

---

> > > ### Author Response · Authors · 2023-08-20
> > > **Looking forward to your reply**
> > >
> > > Hello. The author-reviewer discussion period is going to end. We wonder if our rebuttal solves your concerns. If there remains questions, we are willing to conduct further and deeper discussion with you.

---

### Official Review · Reviewer_8w3X · 2023-07-07

**Soundness:** 3 good
**Presentation:** 3 good
**Contribution:** 3 good
**Rating:** 6
**Confidence:** 4

**Summary:**

This paper introduces an interesting approach that harnesses the capabilities of large language models (LMs) to tackle reinforcement learning (RL) problems. The method involves estimating Q-functions using RL algorithms and providing advice to the LMs about actions with high and low Q-values. The expectation is that LMs will utilize their "remembering" abilities to effectively solve the problem. Empirical evaluation on two language-based reinforcement learning tasks demonstrate the enhanced performance achieved by the proposed method.

**Strengths:**

1. The idea of combining a fixed large language model and a reinforcement learning component estimating action-value functions by presenting experience and advice in prompts is, as far as the reviewer is concerned, interesting.

2. The paper exhibits a well-structured organization, featuring clear and coherent writing that is easily comprehensible. Additionally, the experiments provide compelling evidence of the advantages offered by the proposed method.

**Weaknesses:**

1. An essential assumption underlying the proposed method is that large language models (LMs) possess the capacity to "reason" by effectively recalling all instances of success and failure. Hence, it is crucial to engage in a discussion regarding the limitations of this assumption to establish a comprehensive understanding. Questions arise about the case where the observation space is extensive, and if LMs implicitly learn certain types of Q-functions. While the proposed method is simple yet seemingly effective, delving deeper into the understanding of simple methods often yields intriguing insights.

2. The ablation studies are not sufficient, which would provide valuable insights and address specific questions.

3. The applicability of the proposed method to tasks with observations in natural language is evident. However, it remains unclear how the LMs can be adapted to tackle problems involving image or vector-based observations. Further elucidation on this aspect is necessary to ensure a comprehensive understanding of the proposed method's adaptability and versatility across different observation types.

**Questions:**

1. Please respond to the questions raised in the weakness section.

2. What is the architecture of the Q function and what is the reinforcement learning algorithm for learning the Q function?

3. An ablation study where the LLM does not have the access to the discouraged actions is expected.

4. Another critical issue, unrelated to the academic content of the paper, requires attention. It has been brought to attention that the usage of OpenAI's brand in figures, if without proper permission, is highly inappropriate and may constitute a violation.

**Limitations:**

The authors didn't discuss the limitations of the proposed method. The conclusion section summarizes the contribution of the paper.

---

> ### Author Rebuttal · Authors · 2023-08-07
>
> Thanks for your valuable review and advice.
>
> **About extensive observation space**: Extensive observation space may result
> in a much larger experience memory, which may require more scalable and more
> efficient approaches to store the experiences.  Meanwhile, we refer you to the
> experiments in Sec. 4 in the supplementary.  As shown in the results, the
> performance will improve along with increment of the number of experiences and
> there is a saturation, which may be due to the limited number of exemplars. But
> in a more diverse observation space, the saturation may come later. Besides,
> experience merging, observation shrinking (like in Synapse [1]), and forgetting
> mechanism (like in MemoryBank [2]) should help to control the memory size and
> exploit the experiences more effectively.  These perspectives will be studied
> in our future work.
>
> **About if LLM implicitly learns the certain $Q$ function**: We checked the Q
> values predicted by LLM during the test phase on WebShop. The average absolute
> error is 0.417 (reward in WebShop is between 0 and 1), which indicates that *the
> LLM doesn't really learn the certain $Q$ function*. Nevertheless, as stated in
> global reply 1, the LLM is not expected to predict the accurate $Q$ value.
> Instead, we use the experience memory to implement a lookup-based $Q$ function
> and use RL to learn it. As for the LLM, *it is just expected to speculate which
> action is better (to be encouraged) and which one is worse (to be discouraged),
> as well as which action is the best among multiple encouraged actions*. Maybe
> our introduction to the "output format" in Sec. 3.2 confuse you.  We will
> refine our expression in later revision.
>
> **Ablation study where LLM has no access to the discouraged actions**: We
> supplemented experiments on WikiHow and the results are depicted below.
>
> | Setting         | Avg Reward | Success Rate |
> |:----------------|:---------:|:-------------:|
> | Full Model      |    2.63   |      0.93     |
> | w/o Discouraged |    2.48   |      0.81     |
>
> After removing the discouraged actions in the prompt, the performance seriously
> degrades on WikiHow. We inspected the learned experience memories and the
> exemplars chosen by the agents. It is found that sometimes there are no proper
> actions to encourage in the retrieved experience. *In such cases, the
> discouraged actions will help the LLM to avoid several wrong attempts.  If the
> discouraged actions are then not accessible, the LLM will receive no valuable
> guidance from the experience.* This will lead to a poorer performance. Another
> interesting discovery is that ablation model are more willing to imitate or
> repeat the actions given in the action history in the aforementioned
> circumstances rather than randomly explore. This mechanism is not clear to us
> by now.  We will update these new phenomena and perspectives in later revision.
>
> **Adaptability and versatility across different observation types**: We refer
> you to global reply 2 for this question.
>
> **Architecture and learning algorithm for $Q$ function**: $Q$ function is
> implemented with the experience memory as a lookup-based function. We implement
> a basic Q-Learning algorithm to learn the $Q$ function.
>
> **About the figure and limitations**: Thanks for your kind reminder. We will
> update our figure and add Limitation section in later revision.
>
> * [1] Longtao Zheng, Rundong Wang, Bo An. Synapse: Leveraging Few-Shot
>   Exemplars for Human-Level Computer Control. arXiv:2306.07863.
> * [2] Wanjun Zhong, Lianghong Guo, Qiqi Gao, He Ye, Yanlin Wang. MemoryBank:
>   Enhancing Large Language Models with Long-Term Memory. arXiv:2305.10250.

---

> > ### Comment · Reviewer_8w3X · 2023-08-18
> > **Thanks for you response**
> >
> > The author's response has solved most of my concerns. Thanks a lot.
> >
> > It is interesting to see that the LLM does not learn a accurate Q function but a reasonable preference order. It makes the reviewer curious about what will happen in continuous action spaces as future work. The reviewer is still concerned that the proposed method can not be easily extended to problems with extensive observation spaces. Ratings have been increased accordingly. Remember to update your figures!

---

> > > ### Author Response · Authors · 2023-08-20
> > >
> > > Thanks for your kind advice! It is also intriguing for us to make further investigation into extensive observation space and continuous action space in our future work. And we will update our figure and article according to your advice.

---

### Official Review · Reviewer_2hwN · 2023-07-07

**Soundness:** 3 good
**Presentation:** 3 good
**Contribution:** 3 good
**Rating:** 6
**Confidence:** 4

**Summary:**

The paper proposes REMEMBERER, a novel framework for Large Language Models (LLMs) that employs a persistent experience memory and a Reinforcement Learning with Experience Memory (RLEM) mechanism. This setup aims to enable LLMs to learn from previous interaction experiences in decision-making tasks, improving their policies. The experience memory acts as an external repository, storing experiences from past episodes, which the LLM can exploit without the need for fine-tuning its parameters. The framework, tested on two recent Reinforcement Learning (RL) task sets, demonstrated an improvement in performance over previous state-of-the-art (SOTA) models.


**Strengths:**

The paper's primary strength lies in its innovative approach to RL using LLMs. The authors introduce an agent framework, REMEMBERER, that overcomes the typical limitations of existing approaches, such as the significant cost of fine-tuning LLMs. The proposal of RLEM, which updates the experience memory through RL training, enables the system to evolve its abilities in an efficient manner. Moreover, the empirical results validate the effectiveness of REMEMBERER, with it outperforming the SOTA models on two RL tasks. This paper is in line with a large number of recent papers that show that LLMs can be used to create agents.


**Weaknesses:**

While the results are encouraging, the paper doesn't delve deep enough into the impact of different configurations of the REMEMBERER system. It would be beneficial to explore how varying the size and management of the experience memory as well as the similarity functions affects system performance. In addition, the paper focuses on improvement against SOTA models but doesn't provide a comparative analysis against other similar approaches that utilize memory or experience in RL, such as the very recent Voyager paper (which is too recent to compare against, but an example of these types of papers). A weakness of this approach is that it requires using a reward function, when reward functions might be unavailable or hard to create in difficult computer tasks. In comparison to other RL algorithms, the approach seems constrained to text domains.


**Questions:**

What is x in the subscript O_x in the methods section?


**Limitations:**

In the title, I think it should be “Large Language Models are Semi-Parametric Reinforcement Learning Agents”

“Such an agent turns to be a semi-parametric system that can evolve through RL process” -> “Such an agent is a semi-parametric system that can evolve through an RL process”

Overall, I think the paper can be cleaned up a lot with better English and notation.

---

> ### Author Rebuttal · Authors · 2023-08-07
>
> Thanks for your valuable review and advice.
>
> **Impact of memory size**: In practice, we didn't limit the capacity of the
> experience memory, hence it can accommodate as many experiences as the hardware
> memory allows. To have a perspective on the impact of actual memory size, we
> refer you to the experiments in Sec. 4 in the supplementary. Fig. 3 in the
> supplementary and Fig. 1 in the rebuttal PDF demonstrate the number of new
> experiences and total experiences in the memory after each training epoch,
> respectively. The results in Fig. 2 from the supplementary shows that the
> performance improves while the memory size increases, and there is a
> saturation. This saturation may be attributed to the limited number of active
> exemplars in the input prompt.
>
> **Impact of similarity function**:  We didn't include analysis of similarity
> function, as we do not regard this component as the core of the proposed
> Rememberer framework. We mainly focus on building a general framework assisting
> LLM in exploiting its experience. The core is the experience memory and
> RL-based updating approach. The similarity function, in contrast, depends on
> the specific task domain. However, we agree that the selection of similarity
> function may conduct obvious impact on the final performance. We add ablation
> study according to your advice on WikiHow. The results are shown below.
>
> |Setting|Avg Reward|Success Rate|
> |:--|:-:|:-:|
> |Full Model|2.63|0.93|
> |w/o Task Sim. $f_g$|2.63|0.94|
> |w/o Obsrv. Sim. $f_o$|2.47|0.87|
>
> It is noticed that there are not significant difference between performances of
> the full model and the ablation model without task similarity, while the
> performance degrades if *observation similarity* is removed. This may indicate
> that on these tasks, GPT model benefits more from experiences that have similar
> observations rather than similar instruction patterns. (The task similarity for
> WikiHow is implemented according to the instruction pattern and may be too
> coarse to distinguish different experiences.) In the results, we find that
> without observation similarity, Rememberer fails to adjust the retrieved
> experiences according to the specific observation. *E.g.*, similar instructions to access an article may be emitted
> when the agent is on a search result page or a category
> detail page. The actions to be taken are different according to underlying
> pages the agent is on. In such cases, inappropriate experiences cannot bring
> valuable enough guidance, which leads to a poorer performance. As a conclusion,
> it is worth seeking more appropriate and more effective similarity functions
> for specific task domain. We will update our paper according to the new results
> and perspectives.
>
> **Comparison with other LLM-based agents utilizing memory or experience**: We
> have also noticed that a group of related work has been released **after** the
> submission of NeurIPS (*e.g.*, Voyager [1] and GITM [2] at May 25, and ChatDB
> [3] at Jun 7). Voyager proposes to use codes as actions, and designs a skill
> library to store the successful programs as skills.  GITM stores the past
> successful action sequences in a text memory to assist an LLM planner in future
> planning without further discriminating their values.  Then the stored
> experiences are summarized by LLM to gain deeper insights into planning policy.
> ChatDB leverages relational database to track states in a dynamic process. In
> contrast, we store the experiences in a structured memory and learns their $Q$
> values to filters out the more valuable actions. In this way, we can combine RL
> with LLM and design a semi-parametric RL agent.  Meanwhile, some work dedicated
> to long-term or knowledge-augmented conversational tasks also combines LLM with
> memory (*e.g.*, MemoryBank [4], and Ret-LLM [5]). Here, the external memory is
> usually used to extend the context length the LLM can perceive. We are glad to
> include these comparisons in Related Work in future revision.
>
> As the methods like Voyager and GITM are designed for the specific open-world
> environment MineDojo, it will take some effort to migrate them to our test
> benches and will be difficult for us to introduce new experimental results
> during rebuttal. Nonetheless, we find a number of valuable inspirations from
> these work, *e.g.*, automatic curriculum & code as action from Voyager, and
> automatic goal decomposition & experience summarization from GITM. We are
> working to benefit from these ideas in our future work.
>
> **Generalizability**: We refer you to global reply 2 for this question.
>
> **Symbol $O_x$**: $x$ denotes the index set of the experiences retrieved from
> the memory. We use symbol $x$ because the index set depends on the certain
> state met during interaction and cannot be determined in advance and is deemed
> "unknown". We will update the notations and supplement explanations in later
> revision.
>
> **About advice in "Limitations"**: Thanks for your kind opinion. We will update
> the title and expressions in later revision according to your advice.
>
> * [1] Guanzhi Wang, Yuqi Xie, Yunfan Jiang, Ajay Mandlekar, Chaowei Xiao, Yuke
>   Zhu, Linxi Fan, Anima Anandkumar. Voyager: An Open-Ended Embodied Agent with
>   Large Language Models. arXiv:2305.16291.
> * [2] Xizhou Zhu, Yuntao Chen, Hao Tian, Chenxin Tao, Weijie Su, Chenyu Yang,
>   Gao Huang, Bin Li, Lewei Lu, Xiaogang Wang, Yu Qiao, Zhaoxiang Zhang, Jifeng
>   Dai. Ghost in the Minecraft: Generally Capable Agents for Open-World
>   Environments via Large Language Models with Text-based Knowledge and Memory.
>   arXiv:2305.17144.
> * [3] Chenxu Hu, Jie Fu, Chenzhuang Du, Simian Luo, Junbo Zhao, Hang Zhao.
>   ChatDB: Augmenting LLMs with Databases as Their Symbolic Memory.
>   arXiv:2306.03901.
> * [4] Wanjun Zhong, Lianghong Guo, Qiqi Gao, He Ye, Yanlin Wang. MemoryBank:
>   Enhancing Large Language Models with Long-Term Memory. arXiv:2305.10250.
> * [5] Ali Modarressi, Ayyoob Imani, Mohsen Fayyaz, Hinrich Schütze. RET-LLM:
>   Towards a General Read-Write Memory for Large Language Models.
>   arXiv:2023.14322.

---

> > ### Comment · Reviewer_2hwN · 2023-08-14
> > **Reviewer Response**
> >
> > Thank you for your clarifications, revisions, and ablations. After reading your rebuttal and the other reviews I will keep my score as-is for now.

---

### Author Rebuttal · Authors · 2023-08-07

Thanks to all the reviewers for kind review. We collect all the opinions and
give a reply to several common concerns here.

1. **Role of Q-Learning (and what the $Q$ function is)**: We'd like to further
   clarify our motivation and the role of Q-Learning in our proposed Rememberer
   approach. Our motivation is to make LLM utilize its experience and evolve
   its capability. A simple experience memory cannot label the good or bad
   experiences, thus, we introduce RL to learn the $Q$ values for the
   experiences, so that the good and bad experiences can be discrimianted and
   the memory-augmented system can evolve its ability.  In this system,
   Q-Learning is not directly performed onto LLM.  Instead, it is the
   *experience memory* which RL is applied to so as to avoid directly updating
   LLM parameters. *I.e.*, the experience memory constitutes a lookup-based $Q$
   function.  Actually, We don't expect that LLM can accurately predict the $Q$
   value. *LLM is just expected to be capable of discriminating between better
   and worse actions* with the help of exemplars from the experience memory.
2. **Generalizability**: The introduced RLEM framework and Rememberer approach
   are not intrinsically constrained to text domain. The working domain mainly
   depends on the underlying LLM, while Rememberer can work with any available
   LLM per se. *By replacing with a multimodal LLM like GPT-4 or Flamingo,
   Rememberer can also deal with multimodal observations. Besides, even a pure
   text LLM still has an opportunity to handle other modalities.* *E.g.*, some
   work like Socratic Models [1] leverages captioning model to enable a text
   LLM to handle visual inputs. In Voyager [2], the complex observation from
   Minecraft is summarized into texts. In this paper, WebShop and WikiHow are
   also multimodal tasks in fact. The GUIs from them are represented in a text
   format (See Sec. 1 in supplementary). For demonstration, we are conducting a
   simple experiment with Atari environment.  The results may not come up in
   time, however, we will at least describe our plan to represent an Atari
   observation in text format in later replies.

* [1] Andy Zeng, Maria Attarian, Brian Ichter, Krzysztof Choromanski, Adrian
  Wong, Stefan Welker, Federico Tombari, Aveek Purohit, Michael S. Ryoo, Vikas
  Sindhwani, Johnny Lee, Vincent Vanhoucke, Pete Florence. Socratic Models:
  Composing Zero-Shot Multimodal Reasoning with Language. ICLR 2023.
* [2] Guanzhi Wang, Yuqi Xie, Yunfan Jiang, Ajay Mandlekar, Chaowei Xiao, Yuke
  Zhu, Linxi Fan, Anima Anandkumar. Voyager: An Open-Ended Embodied Agent with
  Large Language Models. arXiv:2305.16291.

---

> ### Author Response · Authors · 2023-08-20
> **Representation of Atari observation in text format**
>
> Hello. It seems that we cannot produce new results on Atari in time and we will
> briefly describe our attempt to represent an Atari [1] observation in text
> here.  Let's pick the environment "Breakout" as an example. States of three
> components are described in the observation: bricks, the ball, and the paddle.
> *E.g.*,
>
> ```
> Bricks: (10, 92), (50, 92), (80, 92), (20, 86), (100, 86)
> Ball:
>   Position: (70, 157)
>   Velocity: 5
>   Direction: up +30
> Paddle: 100
> ```
>
> As the bricks may be too many to present their position, we propose to include
> only a part of them through some heuristics (*e.g.*, the lowest ten bricks).
> Detailed information can be extracted directly from the RAM state to form such
> a representation if "RAM" observation mode is adopted. As for "RGB" mode, a
> simple detection model should help.
>
> This is a trivial plan of how to leverage a text LLM on video environments like
> Atari. To describe such visual-rich observations does require more efforts than
> text-rich observations sometimes, but it is still possible.
>
> * [1] https://www.gymlibrary.dev/environments/atari/breakout/

---

### Decision · Program_Chairs · 2023-09-21

**Decision:**

Accept (poster)

**Comment:**

This paper proposes to combine reinforcement learning with large language models by augmenting language model prompts with Q values.  Several reviewers found the paper difficult to follow but others thought the positives, like its innovative core idea, outweigh the negatives.